# Correlation of NTRK1 Downregulation with Low Levels of Tumor-Infiltrating Immune Cells and Poor Prognosis of Prostate Cancer Revealed by Gene Network Analysis

**DOI:** 10.3390/genes13050840

**Published:** 2022-05-08

**Authors:** Arash Bagherabadi, Amirreza Hooshmand, Nooshin Shekari, Prithvi Singh, Samaneh Zolghadri, Agata Stanek, Ravins Dohare

**Affiliations:** 1Department of Biology, Faculty of Sciences, University of Mohaghegh Ardabili, Ardabil 56199-11367, Iran; a.bagherabadi@student.uma.ac.ir; 2Department of Biology, Jahrom Branch, Islamic Azad University, Jahrom 74147-85318, Iran; amirrezahoushmand66@gmail.com; 3Department of Biology, Faculty of Sciences, Shahid Chamran University of Ahvaz, Ahvaz 61357-83151, Iran; shekarinooshin51@gmail.com; 4Centre for Interdisciplinary Research in Basic Sciences, Jamia Millia Islamia, New Delhi 110025, India; prithvi.mastermind@gmail.com (P.S.); ravinsdohare@gmail.com (R.D.); 5Department and Clinic of Internal Medicine, Angiology and Physical Medicine, Faculty of Medical Sciences in Zabrze, Medical University of Silesia, Batorego 15 St., 41-902 Bytom, Poland

**Keywords:** prostate cancer, systems biology, bioinformatics, gene network analysis, biomarker

## Abstract

Prostate cancer (PCa) is a life-threatening heterogeneous malignancy of the urinary tract. Due to the incidence of prostate cancer and the crucial need to elucidate its molecular mechanisms, we searched for possible prognosis impactful genes in PCa using bioinformatics analysis. A script in R language was used for the identification of Differentially Expressed Genes (DEGs) from the GSE69223 dataset. The gene ontology (GO) of the DEGs and the Kyoto Encyclopedia of Genes and Genomes (KEGG) pathway enrichment analysis were performed. A protein–protein interaction (PPI) network was constructed using the STRING online database to identify hub genes. GEPIA and UALCAN databases were utilized for survival analysis and expression validation, and 990 DEGs (316 upregulated and 674 downregulated) were identified. The GO analysis was enriched mainly in the “collagen-containing extracellular matrix”, and the KEGG pathway analysis was enriched mainly in “focal adhesion”. The downregulation of neurotrophic receptor tyrosine kinase 1 (NTRK1) was associated with a poor prognosis of PCa and had a significant positive correlation with infiltrating levels of immune cells. We acquired a collection of pathways related to primary PCa, and our findings invite the further exploration of NTRK1 as a biomarker for early diagnosis and prognosis, and as a future potential molecular therapeutic target for PCa.

## 1. Introduction

The most common malignancy diagnosed in men worldwide is prostate cancer (PCa) [1]. In the most frequent cancers (2018), this malignancy takes the fourth place among other types of cancers [2]. Mortality rates caused by this heterogeneous disease have been static, but it does not alter the increased chances of prostate cancer by a man in the later ages of his life or a man with a family history of prostate cancer [2,3,4]. Statistics have also shown that many black men suffer from this situation compared to white or Asian men [4]. One of the known possible factors influencing the risks of prostate cancer is dietary habits [5]. Prostate cancer is caused by a number of variables, including smoking, obesity, race/ethnicity, food, age, chemical and radiation exposure, sexually transmitted illnesses, and so on [6]. However, the fundamental shift at the molecular level is the confirmed diagnosis of PCa. The glandular development and expression of luminal differentiation markers androgen receptor (AR) and prostate-specific antigen (PSA) identify most prostatic malignancies that are adenocarcinomas [7]. A blood test and biopsy-based PSA efficiently diagnose prostate cancer in the early stages [8]. This is due to mutations and other changes in the AR or signaling pathways that lead to its increased expression [9], with recent research revealing that the knocking out of the AR gene reduces PCa cell invasion and migration [10]. Although PSA is the most widely used biomarker for prostate cancer, it has low specificity and substantial limitations. Therefore, screening new early diagnostic and prognostic markers for the pathogenesis and prognosis of prostate cancer is essential [11]. The mutations in BRCA1 and BRCA2 can give rise to prostate cancer [12,13,14]. It is not expected that a single gene directs the pathogenesis of the disease, and usually, alterations in the gene expression profile form a pathogenetic network built up from interactions between multiple genes.

The genes with equivalent effects in a pathogenetic network are sited in the same functional portion defined as a module, and they cooperate to fulfill their biological function [15,16,17]. Developments in bioinformatics technologies, such as microarrays, transcriptome sequencing, and proteomics, have provided potential advancements in cancer biomarker research and have been surveyed in several studies on different types of cancer [11,18]. Several studies have identified genes that have a significant role in the onset and development of PCa, such as forkhead box A1 (FoxA1) [19], kallikrein-related peptidase 3 (KLK3) [20], insulin-like growth factor 2 (IGF2) [21], and phosphatase and tensin homolog (PTEN) [22]. The essential genes discovered by the previous research, on the other hand, are quite diverse from one another and have nothing in common, which might be explained by the fact that PCa is a heterogeneous illness in general. We used a single microarray data set of publicly available human primary prostate tissue and screened the differentially expressed genes (DEGs) in the first step. Gene annotation and pathway analysis were performed using gene ontology (GO) enrichment and the Kyoto Encyclopedia of Genes and Genomes (KEGG) pathway analysis. A protein–protein interaction (PPI) network was then constructed to identify the hub genes (HUBs). This study aimed to find an effective prognostic gene in primary prostate cancer among HUBs by bioinformatics analysis and validated findings using the cancer genome atlas (TCGA) data. We used the UALCAN and cBioPortal databases to verify the expression levels and mutational conditions. We also looked into immune cell infiltration utilizing the tumor immune estimation resource (TIMER) database.

## 2. Materials and Methods

### 2.1. Microarray Data Extraction

The GSE69223 CEL files were obtained from the National Center for Biotechnology

Information (NCBI) gene expression omnibus (GEO) database (https://www.ncbi.nlm.nih.gov/geo/, accessed on 20 March 2021) [23]. The data set comprised 30 samples; 15 were primary prostate cancer tissue, and 15 were adjacent normal prostate tissue. The tumor staging for primary prostate cancer was pT2 or pT3. Sequencing was performed using the GPL570 platform (Affymetrix Human Genome U133 Plus 2.0 Array) platform with 54,675 probes.

### 2.2. Data Preprocessing and Screening of DEGs

The unprocessed CEL files were background-corrected, normalized, and converted to an expression set by the “affy” package (https://www.r-project.org/, accessed on 20 March 2021, Version 4.0.5) using the MAS 5.0 expression measure (mas5) function and then $\log_{2}\;$ transformation was applied to the expression values. The expression data were scaled using the scale function in R, followed by applying the principal component analysis (PCA) to remove the outlier samples. The probe IDs were translated into their HUGO Gene Nomenclature Committee (HGNC) gene symbols corresponding to the official sequencing platform. The maximum value across the probes was used to compute the expression value for a particular gene symbol represented by multiple probe IDs [24], and probes that did not have gene information were excluded. Finally, the Escape Excel plug-in [25] was used to prevent gene name mangling using Microsoft Excel (2019). Empirical Bayes statistics (eBayes, *p*-value < 0.05) was applied through the “limma” package [26] in R to uncover the DEGs between normal and cancer tissue, according to the following criteria: $\left|\log_{2}\mathrm{FC}\right|>1.5$ and adjusted *p*-value (FDR) < 0.05. The adjusted *p*-value was calculated using the Benjamini–Hochberg (BH) method by limma package.

### 2.3. Pathway and Functional Enrichment Analysis

The “clusterProfiler” package in R [27] was applied to annotate and visualize the functional profiles for the DEGs. GO term enrichment [28] and KEGG pathway analyses [29] were performed using this package. The cell component (CC), biological process (BP), and molecular function (MF) terms associated with the DEGs were characterized by the GO enrichment analysis. The KEGG pathway analysis uncovered biological pathways correlated with the DEGs. The threshold was set at an adjusted *p*-value < 0.05 (obtained using the BH procedure). 

### 2.4. PPI Network Construction and HUBs Selection

The species *Homo sapiens* was chosen, and the PPI network was constructed using the Search Tool for the Retrieval of Interacting Genes/Proteins (STRING, https://string-db.org/, accessed on 12 May 2021) v11.0 b database [30] to investigate the interactions between the DEGs. Only experimentally validated interactions with the minimum required interaction score 0.4 (default) were selected and retained. Cytoscape (v 3.9.0) [31] was used to analyze the network parameters for further HUBs identification and sub-network visualization. The eigenvector was computed using the “CentiScaPe 2.2” plug-in, while other topological parameters were computed using the Cytoscape network analyzer tool. 

### 2.5. Overall Survival (OS) Analysis of HUBs

In this case, we used the Gene Expression Profiling and Interactive Analysis (GEPIA, http://gepia2.cancer-pku.cn/, accessed on 2 August 2021) v2 database [32], which is an RNA-Seq web server based on the UCSC Xena project, calculated by a standard pipeline. We used the GEPIA to see if the expressions of the hub genes were relevant to the survival of PCa patients in the TCGA cohort. Patients with a high level of expression (>median expression value) and patients with a low level of expression (<median expression value) were defined. The Kaplan–Meier (KM) method was used to evaluate overall survival [33] using a log-rank test (statistically significant: *p*-value < 0.05). 

### 2.6. Validation of Prognostic HUBs Using cBioPortal and UALCAN Databases 

The cBioPortal for Cancer Genomics (https://www.cbioportal.org/, accessed on 23 December 2021) [34] and UALCAN (http://ualcan.path.uab.edu/, accessed on 20 March 2022) [35] was tested for expression validation and investigation of genomic alteration frequencies, including mutations and CNA (amplifications and homozygous deletions) in our prognostic PCa HUBs, respectively. The prostate adenocarcinoma (PRAD) dataset of the TCGA was selected in both web-based tools for analysis. 

### 2.7. Tumor Infiltration Analysis

The tumor immune estimation resource (TIMER) web-based tool (http://timer.cistrome.org/, accessed on 23 December 2021) [36] was queried to explore the correlation between tumor-infiltrating immune cells in PRAD patients and the expression levels of the prognostic PCa HUBs. The Spearman’s test was used, and the *p*-value < 0.05 was regarded as the statistically significant threshold.

## 3. Results

### 3.1. Data Preprocessing and Identification of DEGs 

Because outlying microarray samples can dramatically bias further analysis, preprocessing procedures to identify and eliminate such samples from each dataset prior to network construction were critical [37]. Normalization, background correction, and perfect match (PM)/mismatch (MM) correction were applied using the “mas” algorithm of the “affy” package. Additionally, log_2_ (expression values + 1) was generated. Sample outlier detection (using PCA) was subsequent to the normalization and logarithmic transformation of all the probe sets. Box plots of 30 samples were plotted before and after normalization, which can be seen in Figure 1A,B, respectively. After normalization, each box’s median gene expression values became approximately centric, showing an appropriate normalization and a significant data distribution. Accordingly, no sample with considerable deference in Interquartile Range (IQR) was considered an outlier. In addition, Figure 1C,D depicts the PCA plot for the samples in our dataset before and after the omission of outliers, respectively. Between two groups (tumor and normal), four samples were farther away from their clusters, because of which there was an inherent overlap between the two clusters (tumor and normal), which were considered outliers. On removal of these four samples, we saw that the two clusters were distinct and independent from each other. In conclusion, the outlier samples were: GSM1695588_K400_PN (K400_PN), GSM1695600_K815_PN (K815_PN), GSM1695606_L083_PN (L083_PN), and GSM1695593_K643_PC (K643_PC).

The probe IDs were translated into HGNC gene symbols corresponding to the hgu133plus2.db platform using the annotate R package. Utilizing limma, 990 DEGs were identified in PCa (threshold: $\left|\log_{2}\mathrm{FC}\right|>1.5$ and adjusted *p*-value < 0.05), including 316 upregulated and 674 downregulated genes. Table 1 represents the top ten DEGs that were upregulated and downregulated based on the order of fold changes. Table 2 shows the known PCa biomarkers available within our list of DEGs. The volcano plot was used to visualize the DEGs dispersions (Figure 2A). The expression pattern of the DEGs between samples was revealed by hierarchical clustering analysis, indicating that tumor tissues have substantially different gene expression patterns compared to adjacent normal tissues (Figure 2B). Appendix A provided detailed information on the upregulated and downregulated DEGs.

### 3.2. DEGs Enrichment Analysis

As reported by the KEGG pathway analysis obtained from the clusterProfiler package, the DEGs were linked to pathways, including focal adhesion, dilated cardiomyopathy, protein digestion and absorption, hypertrophic cardiomyopathy, calcium signaling pathway, arrhythmogenic right ventricular cardiomyopathy, ECM-receptor interaction, PI3K-Akt signaling pathway, proteoglycans in cancer, and arginine and proline metabolism (Table 3). The top 20 enriched KEGG pathways are depicted in Figure 3A.

The results of the GO analysis, also performed using the clusterProfiler package, revealed mainly the enrichment of DEG in the collagen-containing extracellular matrix (belonging to the cellular component), the extracellular matrix (a member of the biological process), and the organization of the extracellular structure (part of the biological process) (Table 4). The 10 enriched GO terms for the cellular component, the biological process, and the molecular function are shown in Figure 3B. Appendix A provide comprehensive information on the GO and KEGG enrichment findings.

### 3.3. PPI Network Construction and HUBs Selection

Based on the STRING interaction score > 0.4, our network consisted of 886 nodes and 3579 edges built and visualized by Gephi (Figure 4). Various topological/centrality distributions of the PPI network can be seen in Figure 5. The top 50 genes ranked on the basis of four topological algorithms (i.e., Degree, Betweenness, Closeness, and Eigenvector) were carried out to detect the potential key genes in the PPI network. As shown by the Venn plot in Figure 6, 22 genes were overlapped within four topological algorithms and were considered HUBs in our study. The epidermal growth factor (EGF), transforming growth factor β 3 (TGFB3), brain-derived neurotrophic factor (BDNF), caveolin 1 (CAV1), cadherin 2 (CDH2), collagen type I α 2 chain (COL1A2), decorin (DCN), fibrillin 1 (FBN1), fibroblast growth factor 2 (FGF2), C-X-C motif chemokine ligand 12 (CXCL12), insulin-like growth factor 1 (IGF1), laminin subunit β 1 (LAMB1), matrix metallopeptidase 14 (MMP14), 5′-nucleotidase ecto (NT5E), neurotrophic receptor tyrosine kinase 1 (NTRK1), peroxisome proliferator activated receptor γ (PPARG), epithelial cell adhesion molecule (EPCAM), snail family transcriptional repressor 2 (SNAI2), bone morphogenetic protein 4 (BMP4), neural cell adhesion molecule 1 (NCAM1), secreted protein acidic and cysteine rich (SPARC), and vinculin (VCL) were our hub genes. EGF and EPCAM were upregulated, and others were downregulated. Among the DEGs, EGF had the highest node degree (99). 

### 3.4. OS Analysis of HUBs

We used GEPIA to explore the prognostic worthiness of HUBs and performed an OS analysis. As shown in Figure 7, only NTRK1 (downregulated) had a statistically significant (log-rank *p*-value < 0.05) impact on the PCa patients’ OS. As a result, the relatively low expression of NTRK1 was significantly related to a poor prognosis of PCa, while the remaining 21 hub genes were not reported to be correlated to PCa prognosis (Appendix A).

### 3.5. Validation of Prognostic HUBs Using cBioPortal and UALCAN Databases

The cBioPortal was used to investigate the specific genetic alterations of NTRK1 (prognostically significant) and 21 other hub genes across the TCGA-PRAD messenger RNA (mRNA) cohort comprising 501 patient samples. All these 22 HUBs were altered in 34% (i.e., 172 cases) of the patient samples, most of which were deep deletions (19.24%: altered in 96 cases) followed by amplifications (6.61%: altered in 33 cases), missense mutations (4.61%: altered in 23 cases), and multiple alterations (4.01%: altered in 20 cases). OncoPrint results for all these HUBs, as shown in Appendix A, revealed genetic alterations, including amplification, mRNA upregulation, mRNA downregulation, and various others in 34% of patient samples. The top five highly mutated HUBs were NT5E, SNAI2, FBN1, FGF2, and NTRK1. As only NTRK1 was prognostically significant, unlike other HUBs, we further analyzed only this gene and eliminated the rest. Figure 8A shows the overall alteration frequency barplot of NTRK1 in the PRAD dataset with a deep deletion frequency of 1.8% (9 samples), a missense mutation frequency of 1.2% (6 samples), and an amplification frequency of 0.4% (2 samples). As shown in Figure 8B, the Lollipop plot displayed the frequency and location of all possible mutations in the Pfam protein domains where NTRK1 had a somatic mutation frequency of 1.2% (i.e., eight missense mutations + 1 inframe mutation).

The NTRK1 expression in the TCGA-PRAD cohort was validated using the UALCAN database, based on the sample types, the nodal metastasis, the molecular signature, and the TP53 mutation status. As shown in Figure 8C, the expression level of NTRK1 in the primary tumor and the normal cells is significantly different (*p*-value $=1.73\times 10^{-3}$). In Figure 8D,F, a significant correlation of NTRK1 expression with nodal metastasis [normal vs. N0 (*p*-value $=1.74\times 10^{-3}$), normal vs. N1 (*p*-value $=1.62\times 10^{-3}$)] and TP53 mutation status [normal vs. TP53-Mutant (*p*-value $=6.01\times 10^{-4}$), normal vs.TP53-nonmutant (*p*-value $=2.00\times 10^{-3}$)] is depicted. Figure 8E represents significant differential expression based on molecular signatures [normal NTRK1 vs. ERG fusion (*p*-value $=6.62\times 10^{-4}$), normal vs. ETV1 fusion (*p*-value $=4.22\times 10^{-4}$), normal vs. FLI1 fusion (*p*-value $=2.12\times 10^{-3}$), and normal vs. SPOP mutation (*p*-value $=1.06\times 10^{-3}$)].

### 3.6. Tumor Infiltration Analysis

The TIMER database was used to evaluate the significant correlation of NTRK1 with tumor-infiltrating immune cells across TCGA-PRAD cohort. The NTRK1 expression had a significant positive correlation with the infiltrating levels of T cell CD8+ (r=0.127, *p*-value $=9.71\times 10^{-3}$), T cell CD4+ (r=0.275, *p*-value $=1.17\times 10^{-8}$), dendritic cell (r=0.347, *p*-value $=3.34\times 10^{-13}$), macrophage (r=0.123, *p*-value $=1.18\times 10^{-2}$), neutrophil (r=0.264, *p*-value $=4.47\times 10^{-8}$), and T cell NK (r=0.103, *p*-value $=3.49\times 10^{-2}$), as shown in Figure 9, respectively.

## 4. Discussion

Finding new potential biomarkers to achieve the early detection of PCa in the primary stages and identify new molecular drug targets is necessary and plays a considerable role in helping patients most likely to benefit from treatment. Our study concentrated on a single cohort profile dataset through a microarray analysis. Therefore, we tried to achieve the highest possible quality in terms of gene expression levels by removing the outlier samples and using an efficient normalization method. We identified 990 DEGs between the primary PCa and the adjacent normal tissues, including 316 upregulated and 674 downregulated genes from the GEO dataset-GSE69223. It is apparent that the number of downregulated genes is notably higher than the upregulated genes. We performed a functional enrichment analysis using the clusterProfiler package to better understand the interactions among the DEGs. The DEGs were involved mainly in the organization of the collagen-containing extracellular structure, the organization of the extracellular matrix, the structural constituent of the extracellular matrix, the extracellular matrix, and the sarcolemma, according to the current enrichment analysis of the GO term (Table 4). The enrichment analysis of the KEGG pathway showed that the DEGs were involved in several cancer-related pathways: the calcium signaling pathway, the PI3K-Akt signaling pathway, the Wnt signaling pathway, the cGMP-PKG signaling pathway, and the focal adhesion, which are important in tumor growth and carcinogenesis [38,39,40,41,42,43]. In addition, the DEGs were mainly involved in the digestion of proteins and the absorption and cardiomyopathy pathways (Table 3). Our analysis represents NTRK1 as a new protein involved in PCa prognosis. Moreover, as we have shown in Appendix A (summarized in Table 2), we obtained some known PCa biomarkers such as α-methylacyl-CoA racemase (AMACR) [44,45], forkhead box A1 (FOXA1) [46,47,48], two members of the kallikrein related peptidase (KLK) gene family KLK2 [49,50] and KLK4 [51,52,53], prostate-cancer-associated 3 (PCA3) [54,55,56], distal-less homeobox 1 (DLX1), and a member of the homeobox (*HOX*) family HOXC6 [57], which play specific roles in PCa development.

AMACR plays a key role in the peroxisomal β-oxidation of dietary branched fatty acids. Previous studies have shown that AMACR is upregulated in prostate cancer. However, the mechanism underlying the correlation between AMACR and prostate cancer has not been clarified yet. In a meta-analysis of 22 studies on 4385 participants from various geographic regions, the results show the association between PCa risk and AMACR. In this study, AMACR expression is significantly associated with an increased diagnosis of PCa [58]. FOXA1 helps to shape AR signaling through direct interactions with the AR and drives the growth and survival of normal prostate and prostate cancer cells. FOXA1 also possesses an AR-independent role in regulating epithelial-to-mesenchymal transition [47]. Previous in vitro studies have shown that FOXA1 increases pro-angiogenic factors, including EGF, endothelin-1, and endoglin in prostate cancer cells and promotes endothelial cell proliferation, migration, and tube formation. Moreover, in vivo studies and a clinical samples investigation have shown that FOXA1 facilitates prostate cancer angiogenesis [59].

KLKs are involved in the regulation of tumor growth, neoplastic progression, tumor angiogenesis, and metastasis. Tailor et al. exhibited significant gene upregulation of KLK2 and KLK4 in PRAD. KLK2 can increase ECM degradation due to its proteolytic effects on fibronectin, laminin, gelatin, fibrinogen, and collagenases, leading to metastasis. In addition, KLK4 has been shown to increase the activation of plasmin via the activation of the urokinase plasminogen activator, which helps with the angiogenesis, invasion, and metastasis of the tumor [60].

PCA3 is significantly elevated in patients with prostate cancer, and several available studies show the utility of PCA3, as a urinary biomarker, for the diagnosis of early prostate cancer with reasonable specificity and sensitivity [61,62]. Liang et al. have shown that DLX1 is upregulated in the prostate clinical samples and exerts its oncogenic roles on PCa by activating β-catenin/TCF signaling and promoting the growth and migration of PCa cells [63]. In earlier studies, it has been reported that HOXC6 is involved in PCa development. Recently, Zhou et al. have shown that upregulated HOXC6 could participate in the progression of PCa and function as an independent prognostic marker for cancer [64].

Subsequently, after bioinformatical analysis and the construction of the PPI network, we identified potential HUBs in primary PCa (Figure 2C). One of the key genes with the highest degree (99) in the network was the EGF, which was upregulated in this study, and only NTRK1 (downregulated) was significantly (*p*-value < 0.05) associated with PCa prognosis (Figure 7). A UALCAN-based analysis in PRAD supported that the downregulation of NTRK1 was significantly associated with tumorigenesis (Figure 8C), and this expression decreases when the status of the nodal metastasis increases (Figure 8D). The TP53 mutation status does not affect the NTRK1 downregulation in tumor cells, according to (Figure 8F). As represented in (Figure 8A,B), there was an intermediate tumor mutational burden (TMB) in NTRK1. NTRK1 had a somatic mutation frequency of 1.2%, and most of the mutations are accumulated in the tyrosine kinase domain (Pkinase_Tyr).

NTRK1 or TrkA is a nerve growth factor (NGF) receptor that is part of the tyrosine kinase receptor family. Protein kinases play a critical role in PCa tumor proliferation, development, and metastasis [65], and several malignancies have been shown to cause changes in NTRK1 expression or mutations in this gene [66]. NTRK1 is actively involved in developing, protecting, and maintaining neurons [67,68,69,70]. This study shows the downregulation of NTRK1 in PCa cells, and thus its tumor-suppressive role is expected. However, the tumor-suppressive behavior of NTRK1 in PCa is controversial as many studies have delineated the pro-tumorigenic role of NTRK1 in PCa. The alternative splicing of NTRK1 could result in numerous different protein isoforms, and three of them (TrkAI, TrkAII, and TrkAIII) have been described in humans [66]. The human NTRK1, which is 25 kb in length, is located on chromosome 1q21-q22 and is made up of 17 exons. The full-length isoform is TrkAII, is mostly found in neuronal tissues, and if we remove exon 9 there will be TrkAI in most non-neuronal tissues. Exons 6, 7, and 9 of TrkAIII are missing, making it unable to bind to NGF; therefore, TrkAIII is autophosphorylated, does not bind to NGF, and antagonizes NGF/TrkAI signaling [66,71]. 

As we understood, the oncogenic and tumor-suppressive nature of NTRK1 depends on several things, such as the tumor environment and NTRK1 splicing patterns. Studies have shown that the upregulation of regular NTRK1 isoforms (TrkAI/II) in normoxia condition will have a good prognosis in neuroblastoma (NB), so it plays an antioncogenic role in NB. Moreover, under hypoxic conditions, the upregulation of TrkAIII occurs, which provides tumor progression and metastasis promotion; it plays an oncogenic role in NB and indicates relevance to the NB-regulated tumor-promoting switch by generating an angiogenic, stress-resistant, and tumorigenic NB phenotype via IP/Akt signaling [71,72,73,74,75]. At the molecular level, the activation of NTRK1 confers pro-differentiation programs by binding the specific ligand, binding the NGF, inhibiting angiogenesis, increasing immunogenicity, inducing the differentiation and growth arrest, and mediating apoptosis. On the contrary, downregulating NTRK1 results in proliferation and angiogenesis and thus tumor growth and aggressiveness [73]. Future studies on the splicing patterns of NTRK1 in PCa are needed, and we think that the downregulation of NTRK1 can still be meaningful in the poor prognosis of PCa patients.

According to (Figure 9), it can be inferred that there is a positive correlation between the suppression of the NTRK1 gene and the decrease in immune cell infiltration, such as T cell CD8+. CD8+ T cells and monocytes have been known to suppress PCa. The presence of iNKT cells has been shown to delay prostate cancer progression; however, M1 macrophages and neutrophils infiltration are associated with a poor prognosis [76]. The higher number of T cells, especially CD8+ T cells, memory cells, and CD4+ Th1 cells, can produce a better prognosis in some cancers [77]. Moreover, CD4+ T cells have been linked to human cancers, and they are thought to play a role in PCa growth and promotion [76,78]. As mentioned before, some immune cells such as CD8+ T, monocytes, and NKT cells have a cancer suppression pattern; thus, low levels of these cells’ infiltration could lead to a poor prognosis. However, other immune cells (such as M1 macrophages and neutrophils) have a cancer amplifier pattern. However, these are in line with our findings, which show that intermediate TMB in NTRK1 causes low levels of tumor infiltration of immune cells, notably CD8+ T cells, and the role of CD8+ T cells is more established over other immune cells in the killing of cancerous cells and is used in cancer immunotherapy [79]. The stage of the disease where the host immune response may decrease with increased tumor development has been proposed as a primary factor of immune cell infiltration [80]. Pajtler et al. have provided some evidence that NTRK1 leads to increased immunogenicity in neuroblastoma, which may contribute to a less malignant phenotype and/or the spontaneous regression of neuroblastoma cells [81]. It can be suspected that the suppression of the NTRK1 gene might ignite prostate cancer with the decrease in the level of infiltration of CD8+ T cells.

Finally, in the NTRK fusion-positive tumors, genomic co-alterations and DNA rearrangements were often found, notably in genes implicated in cell-cycle-associated regulators, PI3K signaling, the MAPK pathway, and the tyrosine kinase families [70]. Fusions have been regularly found in rare malignancies, including mammary analog secretory carcinoma, congenital infantile fibrosarcoma, secretory breast carcinoma, and congenital mesoblastic nephroma, as well as in several pediatric case malignancies. NTRK gene fusions can be found in a small percentage of frequent adult cancers, including head and neck cancer, non-small-cell lung cancer, colorectal cancer, salivary gland cancer, thyroid cancer, bladder cancer, brain tumors, and soft tissue sarcomas [66,82,83]. Figure 8E shows a decrease in the expression of the NTRK1- FLI1/ETV1/ERG gene fusions compared to normal tissue. The downregulation of NTRK1 has been detected in several cancers, such as PCa and breast cancer, which lead to tumor progression [84].

Note that (1) this research was entirely based on public databases. As a result, more research is required to confirm the validity of the results; (2) although our objective was to identify a trustworthy candidate associated with the prognosis of primary prostate cancer, there is always the probability of having missed some details.

## 5. Conclusions

Consequently, we recognized the key genes and pathways correlated with the pathogenesis and the prognosis of primary PCa by a bioinformatics analysis. The downregulation of NTRK1 was linked with the poor prognosis of PCa and may be used as a prognostic marker of primary PCa. Nevertheless, NTRK1 expression was shown to significantly correlate with immune cell infiltration levels, such as the CD8+ T cells currently used in cancer immunotherapy. Eventually, further molecular biological studies and clinical research are imperative to confirm these results and their specific functional roles in PCa.

## Figures and Tables

**Figure 1 genes-13-00840-f001:**
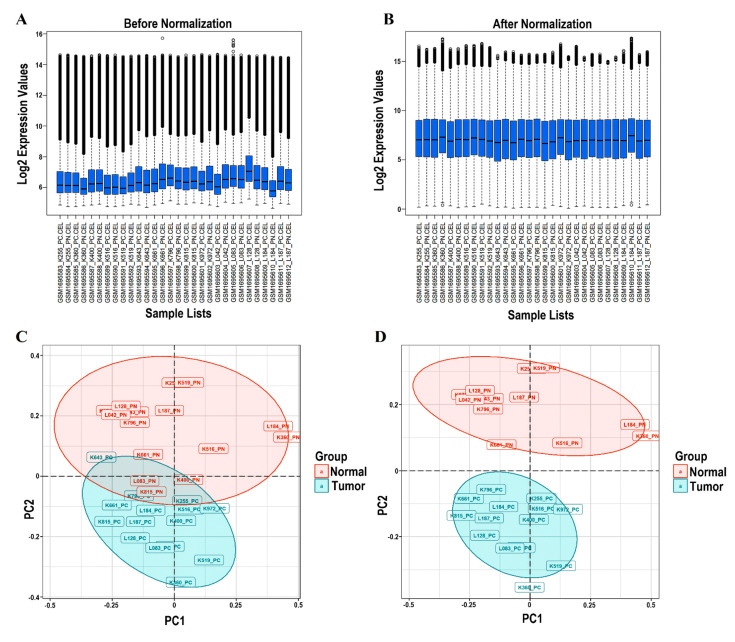
An overview of the preprocessing plots. The box plots show the expression value of the total of 30 samples before and after normalization. The ordinate reflects the gene expression values, while the abscissa displays the distinct samples. (**A**) The median gene expression values within each box (shown by the black line) were not equal before normalization. (**B**) The median of the expression values were nearly on the same line after normalizing, indicating satisfactory normalization performance. (**C**) The PCA plot for 30 samples. (**D**) The PCA plot for the 26 samples selected in this study (K400_PN, K815_PN, L083_PN, and K643_PC excluded). PCA, principal component analysis.

**Figure 2 genes-13-00840-f002:**
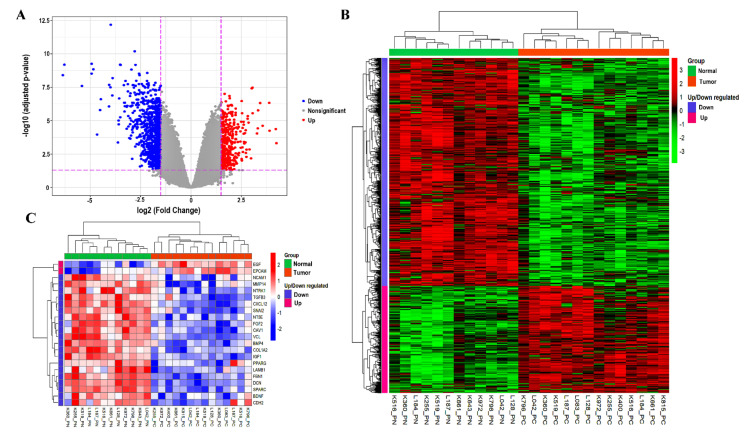
(**A**) The volcano plot highlighting the DEGs. A plot of log_2_FC vs. −log_10_ (adjusted-*p*-value) for the DEGs in PCa shows that the downregulated DEGs highlighted in blue are more than the upregulated DEGs shown in red. The *X*-axis represents the Log_2_FC, whereas the *Y*-axis displays the adjusted *p*-value (−log10 scale). (**B**) The hierarchical clustering analysis of the 990 DEGs. Each column indicates a sample, and each row demonstrates the gene expression level. The color scale ranges from red to green to represent high to low expression. (**C**) The expression heatmap of 22 HUBs. FC, fold change; DEGs, differentially expressed genes; and HUBs, hub genes.

**Figure 3 genes-13-00840-f003:**
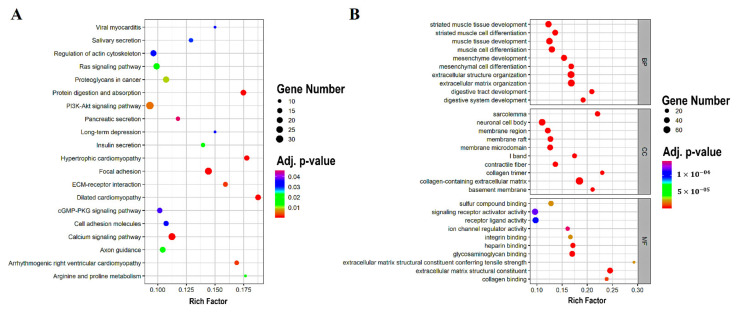
The scatter plots of the KEGG pathway and the GO enrichment. (**A**) The top 20 KEGG pathways are shown in a scatter plot. (**B**) The scatter plot of the top 10 separate GO terms. The Rich factor is the proportion of DEGs numbers among all gene numbers (in the database), indicated in the pathway term that is linked to it. KEGG, Kyoto Encyclopedia of Genes and Genomes; GO, gene ontology; DEGs, differentially expressed genes; BP, biological process; MF, molecular function; and CC, cellular component.

**Figure 4 genes-13-00840-f004:**
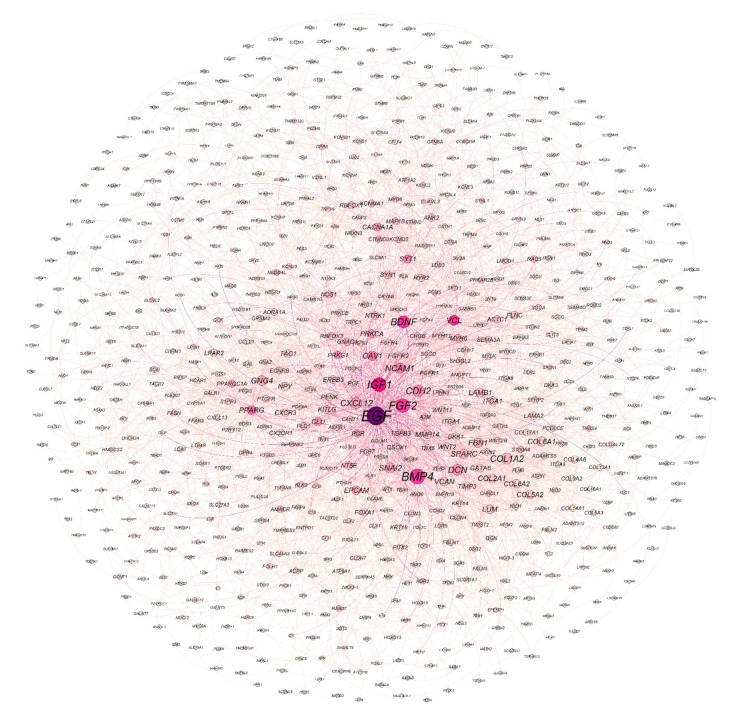
The PPI network of the DEGs. The circle size and color are set for degree and betweenness scores, respectively. The larger the circles, the greater the degree score. In addition, richly colored circles have a higher betweenness score. PPI network, protein–protein interaction network; DEGs, differentially expressed genes.

**Figure 5 genes-13-00840-f005:**
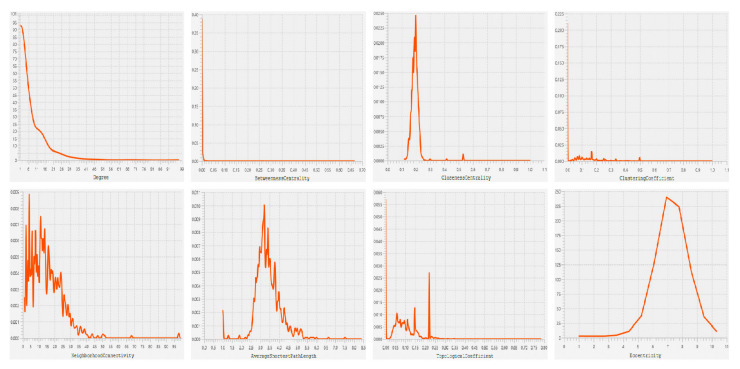
The topological property/centrality distribution plots showing the node degree distribution, the betweenness centrality, the closeness centrality, the clustering coefficient, the neighborhood connectivity, the average shortest path length distribution, the topological coefficient, and the eccentricity of PPI network. PPI network, protein–protein interaction network.

**Figure 6 genes-13-00840-f006:**
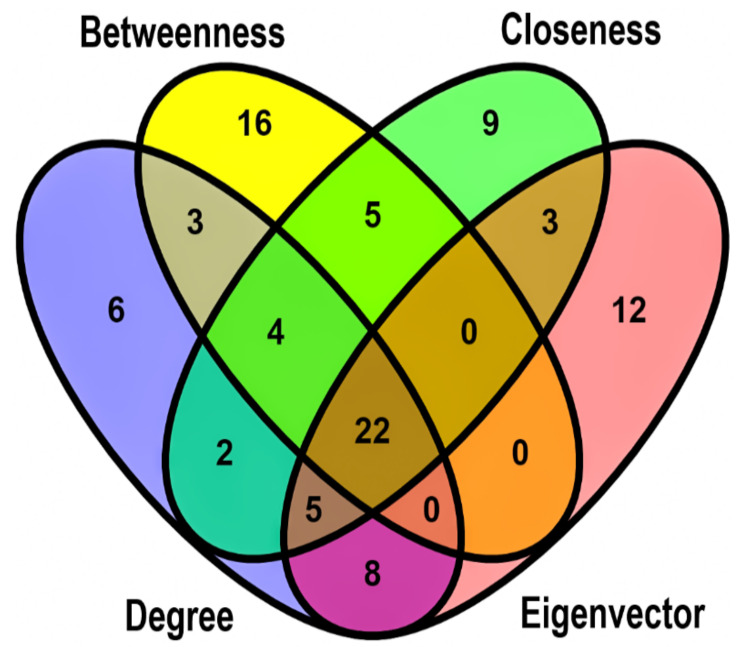
The Venn plot shows 22 overlapping HUBs between the top 50 genes ranked based on four topological algorithms: degree, eigenvector, closeness, and betweenness. The green, yellow, blue, and red areas represent the top 50 genes ranked based on closeness, betweenness, degree, and eigenvector. HUBs, hub genes.

**Figure 7 genes-13-00840-f007:**
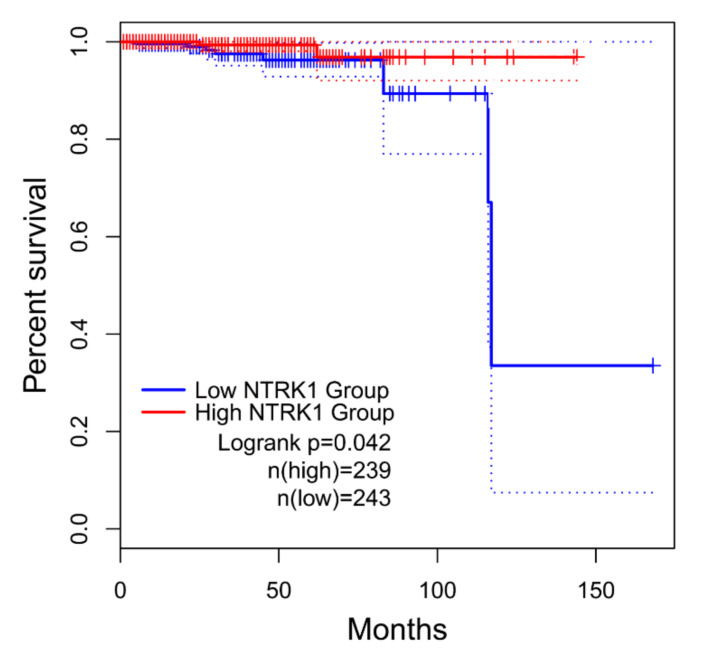
The KM curve was used to estimate OS in PCa patients according to the GEPIA database. The expression of NTRK1 was shown to have a significant impact (log-rank *p*-value < 0.05) on the PCa prognosis using KM estimates (log-rank test). The solid line shows the survival curve, and the dotted line represents the 95% CI. Patients with expression levels above the median are shown with red lines, while those with levels below the median are marked with blue lines. GEPIA, gene expression profiling interactive analysis. OS, overall survival; KM, Kaplan–Meier; PCa, prostate cancer; and CI, confidence interval.

**Figure 8 genes-13-00840-f008:**
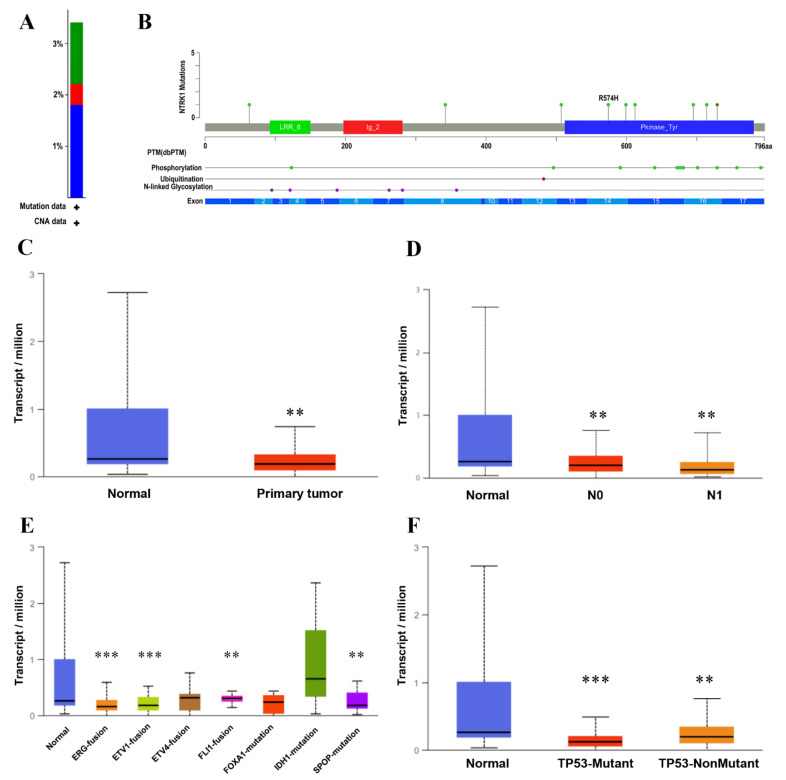
The validation of NTRK1 using the UALCAN and cBioPortal databases. (**A**) The barplot shows the alteration frequency of NTRK1 (3% of 499 PRAD cases) across the TCGA-PRAD dataset (TCGA, Firehose Legacy). The blue bar depicts 1.8% deep deletions, the green bar depicts 1.2% missense mutations, and the red bar depicts 0.4% amplifications. (**B**) The lollipop plot showing nine nonsynonymous mutations in NTRK1 protein domains. The grey horizontal bar represents the whole length of the NTRK1 protein, and the number of amino acids is displayed below the grey bar. The protein domains are shown with the red, blue, and green colored solid boxes. The locations and the frequencies of the mutations were denoted by the solid vertical lines and lollipop-like dots at their ends, respectively. The green and brown lollipops represent eight missense mutations and one inframe mutation. The NTRK1 expression levels in the TCGA-PRAD cohort are shown by box-and-whisker plots based on (**C**) the sample types, and (**D**) the nodal metastasis statuses (N0 means no regional lymph node metastasis; N1 means metastases in 1 to 3 axillary lymph nodes). (**E**) The molecular signatures, and (**F**) the TP53 mutation status via the UALCAN database. ** *p*-value < 0.01, and *** *p*-value < 0.001 vs. normal.

**Figure 9 genes-13-00840-f009:**
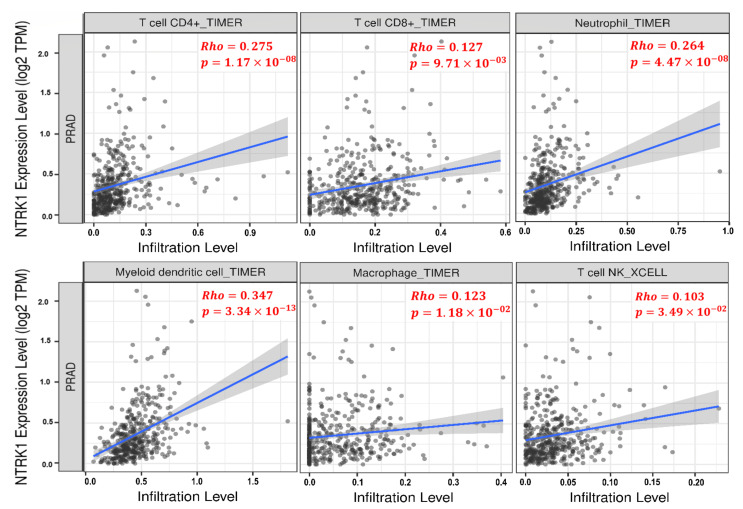
The scatter plots exhibiting significant positive correlations of NTRK1 with the infiltrating levels of T cell CD4+, T cell CD8+, neutrophil, dendritic cell, macrophage, and T cell NK across TCGA-PRAD cohort. In addition, the Spearman’s correlation value and the estimated statistical significance are shown as the legends for each scatter plot. PRAD, prostate adenocarcinoma.

**Table 1 genes-13-00840-t001:** The top 10 DEGs that were upregulated and downregulated between tumor and normal tissues. DEGs, differentially expressed genes; FC, fold change.

Genes Symbol	Probe ID	Log_2_FC	Adjusted *p*-Value	State
OR51E2	236121_at	4.244412	$4.76\times 10^{-4}$	Upregulated
DLX1	242138_at	3.898046	$5.66\times 10^{-5}$	Upregulated
B3GAT1	219521_at	3.883768	$4.70\times 10^{-7}$	Upregulated
LINC00992	239319_at	3.556998	$2.40\times 10^{-5}$	Upregulated
DNASE1	210165_at	3.435794	$4.77\times 10^{-7}$	Upregulated
FFAR2	221345_at	3.332304	$1.67\times 10^{-6}$	Upregulated
FOLH1B	211303_x_at	3.277545	$9.01\times 10^{-5}$	Upregulated
CLDN3	203954_x_at	3.057903	$2.79\times 10^{-6}$	Upregulated
HPN	204934_s_at	3.054792	$3.34\times 10^{-8}$	Upregulated
TRPM4	219360_s_at	3.014302	$3.79\times 10^{-8}$	Upregulated
CXCL13	205242_at	−6.36684	$3.98\times 10^{-9}$	Downregulated
SMTNL2	229730_at	−4.85304	$1.42\times 10^{-9}$	Downregulated
SMR3B	207441_at	−4.66651	$1.09\times 10^{-4}$	Downregulated
FBXL21P	1555412_at	−4.07877	$1.59\times 10^{-6}$	Downregulated
NELL2	203413_at	−3.99192	$6.67\times 10^{-13}$	Downregulated
SERPINA5	209443_at	−3.56837	$2.17\times 10^{-7}$	Downregulated
BMP5	205431_s_at	−3.52423	$6.47\times 10^{-10}$	Downregulated
RBP4	219140_s_at	−3.51651	$5.50\times 10^{-6}$	Downregulated
FOXF2	206377_at	−3.49783	$6.61\times 10^{-10}$	Downregulated
CA3	204865_at	−3.33564	$2.15\times 10^{-8}$	Downregulated

**Table 2 genes-13-00840-t002:** The previously known PCa biomarkers among our DEGs.

Genes Symbol	Probe ID	Log_2_FC	Adjusted *p*-Value	State
AMACR	217111_at	2.272572347	$1.59\times 10^{-2}$	Upregulated
FOXA1	204667_at	1.988720507	$2.92\times 10^{-6}$	Upregulated
KLK2	210339_s_at	1.577097564	$1.51\times 10^{-4}$	Upregulated
KLK4	224062_x_at	1.84883585	$1.64\times 10^{-3}$	Upregulated
PCA3	232575_at	2.979943566	$6.63\times 10^{-3}$	Upregulated
DLX1	242138_at	3.89804597	$5.66\times 10^{-5}$	Upregulated
HOXC6	206858_s_at	2.908075942	$1.42\times 10^{-3}$	Upregulated

**Table 3 genes-13-00840-t003:** The top 10 DEGs enrichment analysis of the KEGG pathway. KEGG, Kyoto Encyclopedia of Genes and Genomes. The Rich factor is the proportion of selected gene numbers (DEGs) compared to all gene numbers involved in each pathway term. The degree of pathway enrichment increases as the Rich factor increases.

KEGG ID	Description	Category	Gene Count	Rich Factor	BH-*p*-Value
has04510	Focal adhesion	KEGG pathway	29	14.42%	$2.06\times 10^{-5}$
has05414	Dilated cardiomyopathy	KEGG pathway	18	18.75%	$6.87\times 10^{-5}$
has04974	Protein digestion and absorption	KEGG pathway	18	17.47%	$1.35\times 10^{-4}$
has05410	Hypertrophic cardiomyopathy	KEGG pathway	16	17.77%	$3.10\times 10^{-4}$
has04020	Calcium signaling pathway	KEGG pathway	27	11.25%	$1.48\times 10^{-3}$
has05412	Arrhythmogenic right ventricular cardiomyopathy	KEGG pathway	13	16.88%	$2.56\times 10^{-3}$
has04512	ECM–receptor interaction	KEGG pathway	14	15.90%	$2.56\times 10^{-3}$
has04151	PI3K-Akt signaling pathway	KEGG pathway	33	9.32%	$5.47\times 10^{-3}$
has05205	Proteoglycans in cancer	KEGG pathway	22	10.73%	$9.18\times 10^{-3}$
has00330	Arginine and proline metabolism	KEGG pathway	9	17.64%	$1.69\times 10^{-2}$

**Table 4 genes-13-00840-t004:** Top 10 GO enrichment analyses of the DEGs. GO, gene ontology.

GO Term	Description	Category	Gene Count	Rich Factor	BH-*p*-Value
GO:0062023	collagen-containing extracellular matrix	CC	75	18.47%	$8.56\times 10^{-24}$
GO:0030198	extracellular matrix organization	BP	62	16.84%	$3.03\times 10^{-16}$
GO:0043062	extracellular structure organization	BP	62	16.80%	$3.03\times 10^{-16}$
GO:0005201	extracellular matrix structural constituent	MF	40	24.53%	$9.28\times 10^{-16}$
GO:0042383	sarcolemma	CC	30	22.05%	$4.84\times 10^{-11}$
GO:0005539	glycosaminoglycan binding	MF	39	17.03%	$4.73\times 10^{-10}$
GO:0060485	mesenchyme development	BP	43	15.41%	$1.91\times 10^{-9}$
GO:0048762	mesenchymal cell differentiation	BP	37	16.81%	$3.84\times 10^{-9}$
GO:0048565	digestive tract development	BP	28	20.89%	$6.63\times 10^{-9}$.
GO:0042692	muscle cell differentiation	BP	50	12.98%	$1.02\times 10^{-8}$

## Data Availability

This study created or analyzed no new data. Data sharing does not apply to this article.

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
