# Peer review of "Correlation of NTRK1 Downregulation with Low Levels of Tumor-Infiltrating Immune Cells and Poor Prognosis of Prostate Cancer Revealed by Gene Network Analysis"

_genes, 2022, doi:10.3390/genes13050840_

Round 1
Reviewer 1 Report
In this study, Bagherabadi et al. perform a bioinformatics analysis of genes and pathways potentially involved in prostate cancer (PCa). Authors use R language to identify Differentially Expressed Genes (DEGs) from the GSE69223 dataset. Then, they apply the Gene Ontology (GO) and Kyoto Encyclopedia of Genes and Genomes (KEGG) pathway enrichment analysis. They construct a protein-protein interaction (PPI) network using the STRING online database to identify the hub genes (HUBs). GEPIA and UALCAN databases were utilized for survival analysis and expression validation. Authors claim they have identified novel biomarkers for early diagnosis, prognosis, and future potential molecular therapeutic target for PCa.
The study has a value in an attempt to identify potential biomarkers of PCa with state-of-the-art bioinformatics methodology. The weakness of the analysis is that it is not convincing for several reasons:
- there is no clear explanation of why only NTRK1 is correlated with prognosis. As shown in Supl,. Fig 1, there are many other genes with similar Kaplan-Meier plots but are considered not significant. There is no convincing argument to exclude all these alternative genes and apparently no effort has been made to analyse further any of these potential candidates.
- the evidence to validate NTRK1 levels as a biomarker for PCa in the TCGA-PRAD mRNA dataset is not supporting the proposed relationship since out of 499 samples, NTRK1 was altered only in 17 (?.4%), including amplifications.
- the proposed relationship of expression levels of NTRK1 and immune cell infiltration is too speculative.
Another intriguing conclusion is the claim for a correlation of PCa with heart disease as well as patients with coronary heart disease. Authors provide no reasonable explanation on this unexpected correlation.
Reviewer 2 Report
One key concern I am having is that the manuscript used a network-based method, but only 1 gene was identified as candidate. I am wondering if known genes associated with prostate cancer prognosis were also identified in one of the reported DEGs. This can be used for additional validation regarding the quality of the selected microarray data set. Additionally, since you investigated multiple genes in your survival analysis, did you correct for multiple-testing in this step?
Also, please discuss more on those 4 removed samples. How did you decide which samples to remove? Given Figure 1, it seems the “L184” and “K360” samples were further away from the cluster on PC1 axis.
In section 3.3, why there were 818 nodes while there were 990 DEGs? Did you perform further filtering steps here?
Some minor issues:
- The term “Rich factor” was first mentioned in Table 2, please explain the term there or in the main text.
- Figure 2B&C, legend, “up” is better showed on top. Figure 2 description stated, “The color scale ranges from red to green to represent low or high expression”, however the red represents up-regulation in figure legend.
- Figure 8A-D, the significance levels are better visible on the figures.
Round 2
Reviewer 1 Report
In the revised manuscript, Bagherabadi et al. have improved their presentation eliminating those highly speculative aspects regarding the potential application of their findings. However there are still some issues to be considered that authors need to address.
1.- Authors could analyze whether additional genes to NRTK1 among the 22 HUB genes are identified in the TCGA-PRAD mRNA dataset. This is relevant considering the low frequency that NRTK1 is mutated in the 499 samples considered for their study.
2.- It would be nice to incorporate the results mentioned in author’s reply regarding the frequency of mutations in the TCGA-PRAD mRNA dataset. As mentioned “KRAS mutations (11%) are less common than PIK3CA mutations (13%). The BRAF oncogene was found mutated in 8% of all cancers, which is only marginally less common than KRAS”. It is not clear if this relates to PaC or to other cancer types.
3.- The new added Table 2 shows the biomarkers previously associated to PaC. This is an interesting information, but authors do not make any comment to this, nor introduce a proper discussion to justify their apparent irrelevant role and why their findings will be a better tool for diagnosis in PaC. A paragraph in the discussion would be essential to support their case.
Author Response
Dear Reviewer 1
Thanks for the valuable comments
Changes in the body text are written in the color font.
Comment 1: Authors could analyze whether additional genes to NRTK1 among the 22 HUB genes are identified in the TCGA-PRAD mRNA dataset. This is relevant considering the low frequency that NRTK1 is mutated in the 499 samples considered for their study.
Authors’ reply: Thanks for your comment. An OncoPrint showing the overall alteration frequencies in all 22 hub genes across TCGA-PRAD has been added as a Supplementary Figure S2. Now the alteration frequency has been increased due to large number of query genes as previously done with only NTRK1. However, for further analysis, we only selected NTRK1 as it was prognostically significant and reliable, unlike others. Rest 21 hub genes were eliminated from further analysis.
Comment 2: It would be nice to incorporate the results mentioned in author’s reply regarding the frequency of mutations in the TCGA-PRAD mRNA dataset. As mentioned, “KRAS mutations (11%) are less common than PIK3CA mutations (13%). The BRAF oncogene was found mutated in 8% of all cancers, which is only marginally less common than KRAS”. It is not clear if this relates to PCa or to other cancer types.
Authors’ reply: “For instance, KRAS mutations (11%) are less common than PIK3CA mutations (13%). The BRAF oncogene was found mutated in 8% of all cancers, which is only marginally less common than KRAS”.
This is discussed in the case of general cancer and not just specifically PCa (doi:10.1038/s41467-021-26213-y).
There are numerous studies stating that there is an overall less mutational burden in the case of PCa.
- PCa possesses a lower mutational burden than many other epithelial tumor types that are not associated with a strong exogenous mutagen. The somatic point mutations are less common in PCa than in most other solid tumors (doi: 1016/j.cell.2015.10.025).
- Primary and metastatic PCa have low mutation rates and recurrent alterations in a small set of genes (doi: 3389/fgene.2020.595657).
- Point mutations occur less commonly in prostate cancer (doi: 1016/j.eururo.2013.05.029)
Comment 3: The new added Table 2 shows the biomarkers previously associated to PaC. This is an interesting information, but authors do not make any comment to this, nor introduce a proper discussion to justify their apparent irrelevant role and why their findings will be a better tool for diagnosis in PaC. A paragraph in the discussion would be essential to support their case.
Authors’ reply: Thanks for your consideration. Three paragraphs have been added to the discussion to support this information as follows:
- “AMACR plays a key role in peroxisomal beta-oxidation of dietary branched fatty acids. Previous studies have shown that AMACR is upregulated in prostate cancer. However, the mechanism underlying the correlation between AMACR and prostate cancer has not been clarified yet. In a meta-analysis of 22 studies on 4,385 participants from various geographic regions, the results show the association between PCa risk and the AMACR. In this study, AMACR expression is significantly associated with an increased diagnosis of PCa. FOXA1 helps to shape androgen receptor (AR) signaling through direct interactions with the AR and drives the growth and survival of normal prostate and prostate cancer cells. FOXA1 also possesses an AR-independent role in regulating epithelial-to-mesenchymal transition. Previous in vitro studies have shown that FOXA1 increases pro-angiogenic factors including EGF, Endothelin-1, and endoglin in prostate cancer cells and promotes endothelial cell proliferation, migration, and tube formation. Moreover, in vivo studies and clinical samples investigation have shown that FOXA1 facilitates prostate cancer angiogenesis.” [Pg15, Ln346-359]
- “KLKs are involved in the regulation of tumor growth, neoplastic progression, tumor angiogenesis, and metastasis. Tailor et al. exhibited significant gene upregulation of KLK2 and KLK4 in prostate adenocarcinoma. KLK2 can increase ECM degradation due to their proteolytic effects on fibronectin, laminin, gelatin, fibrinogen, and collagenases leading to metastasis. In addition, KLK4 has been shown to increase the activation of plasmin via activation of the urokinase plasminogen activator, which helps with angiogenesis, invasion, and metastasis of the tumor.” [Pg15, Ln360-366]
- “PCA3 is significantly elevated in patients with prostate cancer, and several available studies show the utility of PCA3, as a urinary biomarker, for the diagnosis of early prostate cancer with reasonable specificity and sensitivity. Liang et al. have shown that DLX1 is upregulated in the prostate clinical samples and exerts its oncogenic roles on prostate cancer by activating beta-catenin/TCF signaling and promoting the growth and migration of prostate cancer cells. In earlier studies, it has been reported that HOXC6 is involved in PCa development. Recently, Zhou et al. have shown that upregulated HOXC6 could participate in the progression of PCa and function as an independent prognostic marker for cancer.” [Pg15, Ln367-375]
Dear Reviewer 1
Thanks for valuable comments
Changes in the body text are written in color font.
Comment 1: Authors could analyze whether additional genes to NRTK1 among the 22 HUB genes are identified in the TCGA-PRAD mRNA dataset. This is relevant considering the low frequency that NRTK1 is mutated in the 499 samples considered for their study.
Authors’ reply: Thanks for your comment. An OncoPrint showing the overall alteration frequencies in all 22 hub genes across TCGA-PRAD has been added as a Supplementary Figure S2. Now the alteration frequency has been increased due to large number of query genes as previously done with only NTRK1. However, for further analysis, we only selected NTRK1 as it was prognostically significant and reliable, unlike others. Rest 21 hub genes were eliminated from further analysis.
Comment 2: It would be nice to incorporate the results mentioned in author’s reply regarding the frequency of mutations in the TCGA-PRAD mRNA dataset. As mentioned “KRAS mutations (11%) are less common than PIK3CA mutations (13%). The BRAF oncogene was found mutated in 8% of all cancers, which is only marginally less common than KRAS”. It is not clear if this relates to PCa or to other cancer types.
Authors’ reply: “For instance, KRAS mutations (11%) are less common than PIK3CA mutations (13%). The BRAF oncogene was found mutated in 8% of all cancers, which is only marginally less common than KRAS”.
This is discussed in the case of general cancer and not just specifically PCa (doi:10.1038/s41467-021-26213-y).
There are numerous studies stating that there is an overall less mutational burden in the case of PCa.
- PCa possesses a lower mutational burden than many other epithelial tumor types that are not associated with a strong exogenous mutagen. The somatic point mutations are less common in PCa than in most other solid tumors (doi: 1016/j.cell.2015.10.025).
- Primary and metastatic PCa have low mutation rates and recurrent alterations in a small set of genes (doi: 3389/fgene.2020.595657).
- Point mutations occur less commonly in prostate cancer (doi: 1016/j.eururo.2013.05.029)
Comment 3: The new added Table 2 shows the biomarkers previously associated to PaC. This is an interesting information, but authors do not make any comment to this, nor introduce a proper discussion to justify their apparent irrelevant role and why their findings will be a better tool for diagnosis in PaC. A paragraph in the discussion would be essential to support their case.
Authors’ reply: Thanks for your consideration. Three paragraphs have been added to the discussion to support this information as follows:
- “AMACR plays a key role in peroxisomal beta-oxidation of dietary branched fatty acids. Previous studies have shown that AMACR is upregulated in prostate cancer. However, the mechanism underlying the correlation between AMACR and prostate cancer has not been clarified yet. In a meta-analysis of 22 studies on 4,385 participants from various geographic regions, the results show the association between PCa risk and the AMACR. In this study, AMACR expression is significantly associated with an increased diagnosis of PCa. FOXA1 helps to shape androgen receptor (AR) signaling through direct interactions with the AR and drives the growth and survival of normal prostate and prostate cancer cells. FOXA1 also possesses an AR-independent role in regulating epithelial-to-mesenchymal transition. Previous in vitro studies have shown that FOXA1 increases pro-angiogenic factors including EGF, Endothelin-1, and endoglin in prostate cancer cells and promotes endothelial cell proliferation, migration, and tube formation. Moreover, in vivo studies and clinical samples investigation have shown that FOXA1 facilitates prostate cancer angiogenesis.” [Pg15, Ln346-359]
- “KLKs are involved in the regulation of tumor growth, neoplastic progression, tumor angiogenesis, and metastasis. Tailor et al. exhibited significant gene upregulation of KLK2 and KLK4 in prostate adenocarcinoma. KLK2 can increase ECM degradation due to their proteolytic effects on fibronectin, laminin, gelatin, fibrinogen, and collagenases leading to metastasis. In addition, KLK4 has been shown to increase the activation of plasmin via activation of the urokinase plasminogen activator, which helps with angiogenesis, invasion, and metastasis of the tumor.” [Pg15, Ln360-366]
- “PCA3 is significantly elevated in patients with prostate cancer, and several available studies show the utility of PCA3, as a urinary biomarker, for the diagnosis of early prostate cancer with reasonable specificity and sensitivity. Liang et al. have shown that DLX1 is upregulated in the prostate clinical samples and exerts its oncogenic roles on prostate cancer by activating beta-catenin/TCF signaling and promoting the growth and migration of prostate cancer cells. In earlier studies, it has been reported that HOXC6 is involved in PCa development. Recently, Zhou et al. have shown that upregulated HOXC6 could participate in the progression of PCa and function as an independent prognostic marker for cancer.” [Pg15, Ln367-375]

Reviewer 2 Report
The authors have adequately addressed all my concerns in this revision.
Author Response
Thanks for your nice consideration.